# Natural Substances, Probiotics, and Synthetic Agents in the Treatment and Prevention of Honeybee Nosemosis

**DOI:** 10.3390/pathogens11111269

**Published:** 2022-10-31

**Authors:** Magdalena Kunat-Budzyńska, Michał Budzyński, Michał Schulz, Aneta Strachecka, Marek Gancarz, Robert Rusinek, Aneta A. Ptaszyńska

**Affiliations:** 1Department of Immunobiology, Institute of Biological Sciences, Faculty of Biology and Biotechnology, Maria Curie-Skłodowska University, Akademicka 19, 20-033 Lublin, Poland; 2Department of Invertebrate Ecophysiology and Experimental Biology, University of Life Sciences in Lublin, Doświadczalna 50a, 20-280 Lublin, Poland; 3Faculty of Production and Power Engineering, University of Agriculture in Krakow, Balicka 116B, 30-149 Krakow, Poland; 4Institute of Agrophysics Polish Academy of Sciences, Doświadczalna 4, 20-290 Lublin, Poland

**Keywords:** *Apis mellifera*, *Nosema* spp., *Nosema* *apis*, *Nosema ceranae*, *Nosema neumanni*, adaptogenic plant, alcohol, microbiota

## Abstract

Honeybees are important pollinators, but they are continuously exposed to a variety of fungal and bacterial diseases. One of the various diseases affecting honeybees is nosemosis caused by microsporidia from the *Nosema* genus. Honeybees are mainly infected through consumption of infected food or faeces containing *Nosema* spp. spores. Nosemosis causes damage to the middle intestine epithelium, which leads to food absorption disorders and honeybee malnutrition. Fumagillin, i.e., the antibiotic used to treat nosemosis, was withdrawn in 2016 from EU countries. Therefore, researchers have been looking for compounds of both natural and synthetic origin to fight nosemosis. Such compounds should not have a negative impact on bees but is expected to inhibit the disease. Natural compounds tested against nosemosis include, e.g., essential oils (EOs), plant extracts, propolis, and bacterial metabolites, while synthetic substances tested as anti-nosemosis agents are represented by porphyrins, vitamins, antibiotics, phenolic, ascorbic acids, and others. This publication presents an 18-year overview of various studies of a number of natural and synthetic compounds used in the treatment and prevention of nosemosis cited in PubMed, GoogleScholar, and CrossRef.

## 1. Nosemosis

The honeybee *Apis mellifera* Linnaeus (Hymenoptera: Apidae) is an important crop pollinator around the world. It also contributes to the protection of biodiversity of many insect-pollinated plants [1]. Moreover, it provides many food products, including honey, royal jelly, or bee pollen with healing properties [2,3].

The honeybee is a genetically diverse species. The four main evolutionary lines of these insects, distinguished based on morphology and molecular markers, include line A (areas of Africa), line O (Middle East), line C (Central Europe), and line M (Western and Northern Europe) [4,5]. Recently, the number of bee colonies has been decreasing due to the weather conditions, environmental stress, use of pesticides, and various pathogens, including the *Varroa destructor* mite, the *Paenibacillus larvae* bacterium, and microsporidia from the *Nosema* genus attacking honeybees [3,6].

Due to the lack of mitochondria and the presence of 16S SSU rRNA, *Nosema* spp. were first classified as protozoa; however, more detailed research has revealed that these organisms are representatives of the Fungi kingdom [7]. Until recently, there were two causative agents of nosemosis, i.e., *Nosema apis* and *N*. *ceranae* [8,9]; however, the third species, *N*. *neumanni,* was described in *Apis mellifera* in 2017 [10]. *Nosema apis* is a well-known microorganism, which was already described in 1909 by German biologist Zander [8,11]. *Nosema ceranae* was originally detected in *Apis cerana* in Asia in 1996 [12] and soon in *A. mellifera* [13]. Later studies conducted using polymerase chain reaction (PCR) showed that *Nosema ceranae* was present in honeybees around the world in earlier years [14]. Nowadays, *N. neumanni* is only observed in Uganda, where its incidence was found to be much higher than that of the two other *Nosema* species [10]. Nevertheless, since the *N. ceranae* from Taiwan was able to infect honeybees worldwide, the potential threat of *N. neumanni* invasion should be carefully monitored. 

Honeybee workers can be infected by ingestion of redundant spores of *Nosema* spp. which are found in faeces. Faeces of infected bees are rich in undigested sugars; hence, they are readily licked by other bees. Spores present in the middle intestine attack epithelial cells and can cover the whole intestine lumen, thus disturbing its functions [15]. The middle intestine of the honeybee is the focal point of nutrient absorption [16]. Therefore, the production of digestive enzymes and the proper absorption of food compounds in the middle intestine are disturbed [11,16,17,18,19]. The life cycle of *Nosema* spp. is shown in Figure 1.

Fumagillin isolated from the fungus *Aspergillus fumigatus* was an antibiotic used to treat nosemosis [20]. The approved antibiotic preparation was Fumagilin-B, also known as Fumidil-B ^®^, which contained dicyclohexylamine in addition to fumagillin. Since the invention of this antibiotic by Hanson and Eble [21], it has been used to treat infections caused by *N. apis* [22,23]. The mechanism of action of fumagillin is to inhibit the activity of the enzyme methionine aminopeptidase MetAP-2 [24]. Studies have shown that the use of fumagillin at the dose recommended by the manufacturer reduced the intensity or completely eliminated the infection [25,26]. However, the reduction in the therapeutic dose showed that *N. cerenae* was less sensitive to this antibiotic than *N. apis*. As reported by Huang et al. [27], *Nosema ceranae* escapes fumagillin control and can show resistance to this antibiotic. Fumagillin was withdrawn in 2016 by the European Medicines Agency due to the penetration of fumagillin residues into honey and other bee products, which can be dangerous for consumers of honey and bee products [24,28]. Therefore, there is a high demand to search for other ways to treat nosemosis.

## 2. Natural Substances

### 2.1. Essential Oils

Therapies based on natural compounds can contribute to the control of nosemosis development in honeybees. An alternative to synthetic compounds can be essential oils (EOs), i.e., complex mixtures usually extracted from plants using steam distillation and various solvents such as water, ethanol, phenol, and others [29]. EOs are used in apiculture due to their emollient, calming, carminative, antispasmodic, antiseptic, and antimicrobial properties [30,31,32]. 

One of such preparations was Supresor1, a mixture of EOs derived from mint (*Mentha pepper* L.), lemon balm (*Melissa officinalis* L.), coriander (*Coriander sativum* L.), and thyme (*Satureja hortensis* L.) [30]. According to OECD, the product is practically non-toxic [33], and has been classified into toxicity group 5. According to the Hodge and Sterner (OECD) scale, toxicity group 5 is characterised by the LD 50 value of 5000-15,000 mg/kg [33]. The most effective dose of Supresor1 was 5 mL per 1 L of sugar syrup. This dose used in laboratory experiments led to an 80% reduction in the number of *Nosema* spp. spores in comparison to the control group. Furthermore, even higher concentrations of the preparation did not exert side effects on bees [30].

EOs obtained from Chilean acorn *Cryptocarya alba* (Molina) Looser can be used to control nosemosis as well. The dose of 4 μg per bee was not toxic but effective. Furthermore, the crude extract was more effective than the application of chemically obtained single compounds from *C. alba*, i.e., β-phellandrene, eucalyptol, and α-terpineol [20]. Methanolic extracts (2 to 16%) from other Chilean native plants, *Aristotelia chilensis* (Molina) Stuntz and *Ugni molinae* Turcz., which are rich in rutin and myricetin, also gave promising effects. Nosemosis treatment with EOs from these two plants and propolis decreased the *N. ceranae* spore load and prolonged the honeybee lifespan [34]. 

Another herbal mixture whose effectiveness was tested on *Nosema apis*, *Nosema ceranae*, and mixed infections with both was HEOEM (herbal essential oil extract mixture). HEOEM includes extracts from the following plants: *Rumex acetosella* L., *Achillea millefolium* L., *Plantago lanceolata* L., *Salvia officinalis* L., *Thymus vulgaris* L., *Rosmarinus officinalis* L., and *Laurus nobilis* L. Such a herbal mixture was created due to the chemical composition, the content of biologically active compounds, and the biological activity of these plants. The experiments with the use of HEOEM lasted 3 weeks and were carried out in laboratory conditions in standard cages and in an apiary. Özkırım and Küçüközmen [35] determined the effective dose at 500 ul of HEOEM solution in sugar syrup in laboratory conditions and 2000 ul per frame in an apiary. The results revealed the highest decrease in the number of *Nosema apis* and *Nosema ceranae* spores on days 9 and 12 of the experiment. However, the difference in the spore counts between days 9 and 12 was not statistically significant. The researchers recommended using HEOEM every 3 days for a period of at least 15 days. Due to its natural composition, HEOEM can be an alternative to artificial substances used to fight nosemosis [35].

Natural substances do not always have a positive effect on reducing the number of *Nosema* spp. spores. Vetiver oil derived from *Vetiveria zizanoides* (L.) Roberty from the Poaceae family showed no significant properties against nosemosis. Furthermore, it caused a rapid increase in the *Nosema* infection rates between days 19 and 25 of the experiment [36].

Generally, EOs can be natural competitors to fumagillin in nosemosis treatment. However, they do not seem to outweigh fumagillin effectiveness in any type of treatment.

### 2.2. Plant Extracts

Plant extracts were successfully used to treat *N. ceranae* infection, similarly to EOs. Very promising agents tested both in the laboratory and in the apiary were adaptogenic plant extracts, especially from *Eleutherococcus senticosus* (Rupr. et Maxim.) Maxim. roots. These extracts reduced the nosemosis level, prolonged the honeybee lifespan, and can be used in prophylaxis of nosemosis [37,38].

Extract from *Laurus nobilis* L. (laurel) applied at a 1% concentration reduced the *N. ceranae* spore load [39]. After 17 days of treatment, *Artemisia absinthium* L. extract inhibited development of *N. apis* in both naturally and artificially infected worker honeybees in laboratory conditions. However, it is worth noting that honeybee mortality was also increased [40]. Only extracts from two representatives of the *Compositae* family, *Artemisia dubia* (Wall.) and *Aster scaber* (Thunb.) Nees, exerted an anti-nosemosis effect [41]. Other tested extracts did not show anti-nosemosis activity. These inefficient extracts were derived from *Amaranthus mango-stanys* L., *Mentha arvensis* L., *Allium senescens* L. var. *senescens*, *Astilboides tabularis* (Hemsl.) Engl., *Veratrum oxysepalum* Turcz., *Achyranthes japonica* (Miq.) Nakai, *Lythrum salicaria* L., *Symphytum officinale* L., *Schisandra chinensis* (Turcz.) Baill., *Perilla frutescens* var. *acuta* Kudo, *Physalis alkekengi* var. *francheti* (Mast.) Hort, *Rheum undulatum* L., *Aster scaber* Thunberg, *Cirsium nipponicum* (Maxim.) Makino, *Achillea alpina* (Ledeb), *Disporum uniflorum* Baker, *Astragalus membranaceus* Bunge var. *membranaceus*, *Aster tataricus* L.f., and *Artemisia dubia* (Wall.). 

A decoction of a Chinese herb *Andrographis paniculat* (Burm.f.) Nees administered at a 1% concentration supported epithelium tissue regeneration during *N. ceranae* infection and reduced the number of *Nosema* spores. Other herbs used in this experiment, i.e., *Cyrtomium fortune* J. Sm., *Cinnamomum cassia* (L.) J.Presl, and *Eucalyptus citridora* (Hook.) K.D. Hill & L.A.S.Johnson, seemed to be useless for controlling *N. ceranae* infection, as they increased honeybee mortality, compared to the control group, in laboratory conditions [42]. *Allium sativum* L. (garlic) extract applied at a 1% concentration produced no differences in comparison with the control group. On the other hand, its higher concentration (10%) was highly toxic to honeybees [38]. In turn, *Origanum vulgare* L. and *Rosmarinus officinalis* L. extracts administered at a concentration of 0.7%, and volatile oils exhibited anti-nosemosis properties by reducing spore loads in laboratory conditions [43].

Another plant with activity against nosemosis is *Lespedeza cuneata*. It is an invasive plant displacing native species in many countries and occupying their habitats. The aim of the laboratory experiments conducted by Song et al. [44] was to test the effect of *L. cuneata* extract used at concentrations from 12.5 µg/mL to 800 µg/mL against *N. ceranae* using the *Trichoplusia ni* cell line BTI-TN5B1-4, which is an alternative cell line to honeybee cells. Infected cells were degraded and had a larger, abnormal shape compared to healthy cells with a round and correct shape. When treated with the *L. cuneata* extract, the infected cells were similar in shape to the healthy cells, and most of the spores were outside the cell. In addition, the researchers were able to determine the lowest concentration that inhibited the development of nosemosis, i.e., 50 µg/mL, and the highest concentration: 200 µg/mL, which did not cause adverse effects [44].

In laboratory experiments conducted by Braglia et al. [44], the inhibitory effect of, e.g., *Opuntia ficus-indica* extracts against the development of nosemosis was checked. The extract was administered to honeybees at a concentration of 0.005 µL/mL of sugar syrup (1:1 *w:v*). The study showed that the extract enhanced the development of nosemosis and was toxic to honeybees, as evidence by the death of all honeybees by the 9th day of the experiment [45].

### 2.3. Thymol

Another compound studied in a laboratory was thymol, which reduced the development of *N. ceranae* at a concentration of 0.1 mg per 1 g of honeybee candy or sugar syrup. Similar results can be found in other studies, for example conducted by Yucel and Dogaroglu [46], Maistrello et al. [36], or Borges et al. [47]. Honeybees fed with thymol (0.12 mg/g) and resveratrol (0.001 mg/g) candies exhibited a lower *N. ceranae* infection level [47]. Additionally, honeybees fed with thymol or resveratrol syrup lived longer than those in the control group. However, pure resveratrol did not decrease the *N. ceranae* spore load [47,48].

In apiary experiments conducted by Vargas-Valero et al. [49], *Nosema ceranae*-infected honeybee colonies kept in the tropical conditions of Yucatan, Mexico, were administered a thymol solution at a concentration of 66 mg of thymol crystals per 1 L of sugar syrup or fumagillin at a concentration of 25.2 mg per 1 L of sugar syrup. The administration of the thymol solution proved to be effective in 31.1%, compared to fumagillin, whose effectiveness was 95.2%. In the control group, the nosemosis level decreased after 4 weeks, but this may have been a result of seasonal variations in the nosemosis level. Therefore, further trials using thymol as an alternative antifungal agent are needed [49]. 

### 2.4. Natural Polysaccharide

Chitosan is a linear polysaccharide obtained by deacetylation of chitin. Due to its beneficial properties (e.g., biocompatibility, biodegradability, hydrophilicity, nontoxicity, high bioavailability), it seems to be a safe and promising anti-nosemosis compound [50]. Sugar syrup containing 0.01% chitosan decreased the severity of infection with *N. apis* and increased the expression of AMP and *vitellogenin* genes in apiary conditions [1,47]. Additionally, chitosan prolonged the honeybee lifespan [51,52]. 

### 2.5. Honeybee Product—Propolis

Propolis has many beneficial properties and may also be helpful in treatment of honeybees. It composition depends on plant species available for forager honeybees [53]. Propolis contains many compounds, e.g., flavonoids, aromatic acids, esters, aldehydes, ketones, fatty acids, terpenes, steroids, amino acids, polysaccharides, hydrocarbons, alcohols, hydroxybenzene, caffeic acid, ferulic acid, ellagic acid, quercetin, and others [53]. The propolis ethanol extract administered orally to *Nosema*-infected honeybees significantly reduced the nosemosis level in laboratory conditions [54,55,56]. Furthermore, propolis produced by *Trigona apicalis* Smith, one of the stingless bee species, significantly reduced the *N. ceranae* infection level and mortality in *Apis florea* and *Apis cerana* colonies [53,57,58]. However, propolis preparations also pose some threats to honeybees, as they can be a source of pesticide contamination [54,56].

### 2.6. Probiotics

The first laboratory experiments using gut bacteria isolated from the intestines of healthy honeybees, i.e., bifidobacteria and lactobacilii, were carried out by Baffoni et al. [59]. Laboratory tests using the real-time PCR method proved that the nosemosis level in healthy and infected honeybees fed with gut bacteria was significantly lower than in the control, i.e., honeybees fed only with sugar syrup. Subsequent studies showed that the oral administration of bacterial metabolites produced by *Lactobacillus johnsonii* and *L. kunkeei* strains did not have a toxic effect on honeybees but reduced the nosemosis level [60,61]. Similar results can be found in a study conducted by Audisio et al. [54], in which honeybee colonies were fed *Lactobacillus johnsonii* CRL1647 in the amount of 1 × 10^5^ cfu/mL every 15 days and monthly. The study showed that both methods of administration of the preparation increased honey production and reduced nosemosis. In the case of varroosis, it was observed that administration of lactobacilli once a month reduced the development of the disease [62].

Additionally, organic acids produced by *Lactobacillus johnsonii* CRL 1647 had a positive effect on honeybee colonies and reduced nosemosis development. Additionally, in vitro administration of cell-free supernatants from *L*. *johnsonii* containing mainly organic acids such as lactic acid, phenyl-lactic acid, and acetic acid did not cause honeybee mortality even at a high dose of 60 µL/ honeybee [63]. Furthermore, such metabolites as bactericin and surfactin derived from *Bacillus* strains inhibited the development of nosemosis [64].

On the other hand, honeybees fed with *Lactobacillus rhamnosus* showed a higher nosemosis level and a shorten lifespan in comparison to the control group fed only sugar syrup [65]. 

Therefore, it should be emphasised that supplementation of honeybees’ diet with improperly selected bacteria not dedicated for honeybees does not prevent nosemosis development but may increase its level, deregulate insect immune systems, and significantly increase honeybee mortality [65,66]. 

A metagenomic analysis of honeybee colonies from the UK, Spain, Poland, Greece, and Thailand proved that nosemosis caused higher loads of fungi and such bacterial groups as Firmicutes (*Lactobacillus*), γ-proteobacteria, and Neisseriaceae, whereas healthy honeybees had a higher load of such bacterial groups as Orbales, Gilliamella, Snodgrassella, and Enterobacteriaceae [67,68]. Nosemosis infection also caused a higher abundance of *Bifidobacterium* spp. in infected honeybees [69].

### 2.7. Fungal Extract 

Fungal extracts obtained from the *Agaricus* genus were also used in the control of nosemosis due to their content of biologically active compounds, e.g., glucans, mannan, and lentinan with immunostimulatory effects [70]. In laboratory experiments carried out by Glavinic et al. [71], the influence of aqueous extracts of *Agaricus bisporus* on honeybee survival, the degree of nosemosis infection, and the level of expression of genes related to honeybee immunity, i.e., abacin, defensin, hymenoptecin, apidicin, and vitellogenin, was determined. The concentration of 200 µg/g was selected for further analyses due to its lowest toxicity to honeybees. The researchers observed a decrease in the number of *N. ceranae* spores when the *A. bisporus* extract was administered to the honeybees. Moreover, the infected honeybees had a prolonged lifespan [71]. 

### 2.8. Other Natural Substances

Aqueous extracts of jet-black ant *Lasius fuliginosus* nests in laboratory experiments decreased the nosemosis level. It was found that the administration of carton extract (birch) at concentrations of 0.1% and 1% for 6 days caused an 18-fold decrease in *Nosema* spp. spores, compared to the control group, i.e., honeybees fed only with sugar syrup. Moreover, the extract had a protective effect on healthy honeybees by increasing their lifespan [72].

## 3. Synthetic Substances

### 3.1. Phytochemicals

In addition to substances of natural origin, many chemically synthesised substances were used for nosemosis treatment [73,74]. Bernklau et al. [74] conducted laboratory experiments on honeybees using phytochemicals, i.e., caffeic, gallic, and p-coumaric acids and kaempferol. Caffeic acid at 25 ppm, gallic acid at 250 ppm, and kaempferol at 2500 ppm prolonged the honeybee lifespan most effectively. Caffeic acid at the highest concentration of 2500 ppm reduced the honeybee lifespan. A reduction in the *N. ceranae* spore load was observed during simultaneous administration of caffeic and p-coumaric acids and kaempferol at the concentration of 25 ppm. However, pure kaempferol reduced the amount of spores in all concentrations [74]. Braglia et al. [17] showed in laboratory conditions that p-coumaric acid at a concentration of 31 ppm inhibited the development of nosemosis in winter honeybees. In turn, in the study conducted by Bernklau et al. [74], this acid at a concentration of 25 ppm limited the development of the disease in summer honeybees.

Another tested compound was sulforaphane obtained from cruciferous vegetables. At concentrations of 0.1250 mg/mL and 0.1667 mg/mL, it caused a substantial decrease in the *N. ceranae* spore load. At a concentration of 1.2500 mg/mL, it caused higher honeybee mortality but eradicated nosemosis [47]. Additionally, carvacrol from oregano oil effectively reduced *Nosema* spore loads in a dose of 0.1000 mg/mL of sugar syrup. In turn, naringenin from citrus fruit had a moderate effect on reducing *Nosema* spore loads, and its main advantage was the extension of the honeybee lifespan [47]. 

Caffeine is one of the purine alkaloids naturally occurring in plant species from the *Camellia* L., *Coffea* L., *Theobroma* L., *Paullinia* Kunth, *Ilex* L., and *Cola* H. W. Schott et Endlicher genera [75]. A sugar solution of caffeine administered at a 5 μg/mL dose exerted a protective effect against *Nosema*-infection. Additionally, honeybees fed with a caffeine solution were found to live longer than honeybees fed pure sugar syrup in the control group [76], likewise workers treated with curcumin [77] and piperine [78].

Nicotine is not an effective anti-nosemosis substance and, in a dose of 1 ppm, was not preferentially consumed by honeybee foragers. Higher concentrations of nicotine up to 10^4^ ppm also showed no healing properties. At high concentrations, nicotine increased honeybee mortality. In vitro studies showed that spores treated with nicotine remained infectious to honeybees [79].

### 3.2. Organic Compounds

Other compounds tested to eliminate nosemosis were oxalic, formic, and abscisic acids, which are commonly used by beekeepers to control another disease affecting honeybees, i.e., varroosis caused by *Varroa destructor* [80,81]. After spraying *Nosema*-infected honeybees with oxalic acid, a reduction in the nosemosis level was observed [80,82]. Fumigation with formic acid also decreased *N. apis* and *N. ceranae* spore loads during overwintering [81,83]. The effect of abscisic acid (ABA) applied at a concentration of 50 μM in sugar syrup or honey preserved the initial adult worker population after the winter and, to a small extent, decreased the nosemosis level. Additionally, in both sugar-fed and honey-fed honeybees, much slower development of the disease was observed, compared to control groups that did not receive ABA in apiary conditions [3]. 

A similar experiment was carried out by researchers from the University of Uppsala, who tested the impact of 0.2% and 0.4% acetic acid and 0.03% benzoic acid on the development of nosemosis in laboratory and natural (apiary) conditions. In this experiment, it was shown that the administration of a 10 μL solution of these compounds in sugar syrup did not affect the spread and development of the *Nosema* disease [84]. 

Porphyrins are promising agents in combating nosemosis. These aromatic heterocyclic compounds were another group of compounds used in honeybee supplementation to suppress the development of nosemosis. Honeybees treated with sugar syrup containing synthetic amphiphilic protoporphyrin amide [PP (Asp) 2] had a decreased number of *Nosema* spp. spore loads in laboratory conditions. PP (Asp) 2 has also been shown to reduce the mortality of infected honeybees. Furthermore, incubation of *Nosema* spores with the porphyrin reduced their infection effectivity [85]. In the future, porphyrins may play the role of antibiotics, as they show activity against many microorganisms, e.g., fungi, protozoa, bacteria, and viruses [86].

### 3.3. Alcohol

Another chemical compound tested in laboratory conditions was ethanol. Surprisingly, the amount of *Nosema* spp. spores in infected honeybees treated with 5% ethanol increased significantly compared to the control group and caused higher honeybee mortality [87]. In addition, the administration of 10% ethanol in sugar syrup had a toxic effect on both healthy and *Nosema*-infected honeybees. The administration of this compound probably caused immune suppression in the honeybees, which consequently led to an increase in the nosemosis level. Ethanol may also enable *Nosema* spores to enter the honeybee intestine and facilitate the infection [87]. 

### 3.4. Commercial Preparations 

Another substance obtained synthetically is PROVIGOR 14 WA Bee Care^®^. Its effectiveness is related to the content of bioflavonoids combined with many natural acids, e.g., lactic and ascorbic acids. Honeybee colonies were sprayed twice a week with sugar syrup containing 1 mL of PROVIGOR per 1000 mL of syrup, and a significant reduction in the *Nosema* spore load was observed after 10 such doses. However, the disease was not completely eradicated [88].

The effect of amprolium hydrochloride on *Nosema* disease development was also investigated. It is a veterinary medicine used as an antiprotozoal agent (coccidiostatic). This drug interferes with thiamine metabolism and has a similar structure to vitamin B12. Amprolium hydrochloride was administered to honeybees at two concentrations: 200 ppm and 1000 ppm. The experiment was conducted for 19 days and, at the end, no spores were found in the experimental groups, while the number of spores in the control group ranged from 5.75 to 7.625 million per honeybee [89]. Both concentrations proved to be effective in fighting *Nosema* infection and exerted no side effects on the honeybees. According to the manufacturer’s instructions, the product should be used in the amount of 10 mg/kg of body weight for 7 days in 200 mL of sugar syrup. This dose, even after exceeding the estimated therapeutic dose five times, did not produce side effects in the honeybees. However, the time of treatment should not exceed 5–7 days because this substance may cause vitamin B1 deficiency [89].

### 3.5. Vitamins

A similar experiment to that with amprolium hydrochloride was carried out using B-group vitamins. In the experiment, the following parameters were checked: worker productivity, flight activity, pollen foraging efficiency, effect on egg laying, impact on honey yield, level of parasitation, and others. After 15 days, the condition of the honeybee colonies improved, but there was no effect on the *Nosema* spp. spore load [90].

### 3.6. Antibiotics

Various substances were tested on the IPL-LD 65Y cell line derived from *Lymantria dispar*, which was found to be sensitive to infection with *Nosema* spp. Cell cultures infected with *Nosema* spp. spores were treated with fumagillin as a positive control and with the tested substances, i.e., surfactin, metronidazole, ornidazole, quinine, tinidazole, albendazole, and dimethyl sulfoxide (DMSO). The results of the laboratory experiment showed that antibiotics from the nitroimidazole group, i.e., metronidazole and tinidazole, inhibited the proliferation of *N. ceranae*. However, the other substances did not affect the development of the disease [91].

### 3.7. RNA Interference

RNA interference (RNAi) is used in anti-nosemosis therapy as well. RNAi is a post transcriptional gene silencing mechanism and a natural anti-infective mechanism of the honeybee immune response [92,93]. When workers ingested daily doses of synthetic dsRNA (in sugar syrup) specific to *N. ceranae* ATP/ADP transporters, the target transcript levels and the *Nosema* spore loads decreased [81]. RNAi is also used to lower the expression of polar tube protein 3 (ptp3), i.e., a protein essential for sporoplasm injection and microsporidian cellular invasion [93]. When the expression of ptp3 was decreased, spore loads decreased, several antimicrobial peptides (abaecin, apidaecin, hymenoptaecin, defensin-1) increased in the hemolymph of workers, and the honeybee lifespan was significantly prolonged [81,93]. Following the ingestion of dsRNA targeting the naked cuticle gene (nkd; regulator of immune function in honeybees), lower infection levels and increased immune expression and survival in honeybees were observed [94]. When administering RNAi to honeybees, the action of digestive enzymes and intestinal pH should be taken into account, as they can rapidly metabolize the drug and change its sequence, thereby reducing the efficiency and stability of RNAi [81].

## 4. Conclusions

Similar to other pollinators, honeybees are in danger of extinction [1]. Nosemosis is a very dangerous honeybee disease affecting honeybee physiology and biology. It negatively alters the gut epithelium renewal rate. Mature spores create a layer of spores on the intestine surface, which affects midgut integrity and deteriorates the physiological function of the honeybee alimentary tract [15,95,96]. Furthermore, *Nosema*–infection changes the content of macro- and microelements in honeybees, which disrupts honeybee physiology [15,97]. All these factors lead to higher mortality rates and a decline in honeybee populations. Among the natural substances presented in this review paper, the essential oils of adaptogenic plant extracts, Supresor1, and propolis had the best effect in limiting the development of nosemosis. Among the synthetic substances, promising treatment outcomes were obtained with the use of porphyrins, caffeic acid, and kaempferol administered at appropriate concentrations. Preparations based on synthetic substances have a strictly defined composition, compared to natural substances, whose composition may vary depending on the plant growth locality or the plant collection season. Regardless of the origin of the substance, the key factor is the correct determination of the minimum dose that shows the highest level of inhibition of nosemosis. In addition, an important issue is to determine the duration of the treatment and the frequency of administration of these substances [36]. An excessive dose of the compound very often leads to an increase in honeybee mortality, e.g., administration of 10% garlic extract and 5% ethanol increased honeybee mortality very rapidly. Moreover, the tested compounds should not penetrate and accumulate in honeybee products because this may lead to a change in the flavour of these products and, in the worst case, pose a risk to consumers. Therefore, searching for new agents to prevent and treat nosemosis is an urgent issue, as it is essential to help the honeybee survive. 

## Figures and Tables

**Figure 1 pathogens-11-01269-f001:**
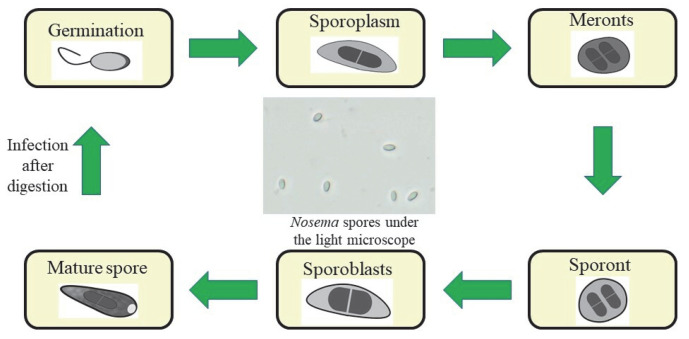
Life cycle of *Nosema* spp. Infection with nosemosis occurs through the ingestion of spores. In the intestine, during germination the spore discharges the polar tube, through which the infected sporoplasm is injected into the cytoplasm of the midgut epithelial cells. The sporoplasm then increases its volume and transforms into a meront. The next stage of development is the stage of sporogony, which is the transformation of meronts into sporonts surrounded by the cell membrane. Next, the sporonts divide into two sporoblasts, which in turn mature into spores. The spores germinate and infect other cells in the intestine of the same honeybee or are expelled outside with faeces and can infect other honeybees.

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
