# Peer review of "Natural Substances, Probiotics, and Synthetic Agents in the Treatment and Prevention of Honeybee Nosemosis"

_pathogens, 2022, doi:10.3390/pathogens11111269_

Round 1

Reviewer 1 Report

The manuscript is not totally original since there are othter published reviews about the topic. However, in the present review, there are some new citations, and the information presented includes valuable details on procedures and results that can be useful to compare experimental approaches.

Please check the comments in the attached file

Author Response

Thank you very much for your insightful remarks. Thank you for your time and effort. We made our best to adhere to your suggestions. The latest version of manuscript was proofread by English language author.

The specific gap is not well presented for readers to get the update by the Content and have a structured understanding on up to date information on mitigation with Nosemosis. Compared with other published material, no addition was achieved. It would be nice to support with info graphs and mode of action details that may inspire applications. The links are missing with the sections and the conclusion. 

We rearranged some paragraph throughout the text to be more specific and easy to follow by readers. We also add some text to be more specific and link sections.

Not well presented using tables and figures. The proposed graphical abstract is not sufficient and relevant. The honey bee nosema life cycle has to be presented, and the mode of action-based interactions of the Natural and synthetic substances mentioned in the review in which stage of the life cycle stage has to be presented with references.

We added the Nosema spp. life cycle scheme and add information about the mode of action of the natural and synthetic substances. We have changed graphical abstract.

However, to give more significance to the review, I would increase the introduction by underlining also the importance of the disease, by further describing the epidemiological relevance of Nosema spp., pathogenesis and alterations of the disease.

We have increase the introduction by underlining the importance of nosemosis and describe  pathogenesis and alterations of the disease. We have also added the Nosema spp. life cycle scheme.

Additionally, we have addressed all specific comments added to the manuscript and discussed with all suggested publication in the field.

We made our best to adhere to your suggestions. After all improvement you have suggested the article is much better and easier to follow by readers.

Thank you once more,

Authors

Reviewer 2 Report

I appreciate the effort of the authors of the review. 

In this review, the authors try to summarize the literature on Natural and synthetic substances in the treatment and prevention of honeybee nosemosis. 

The specific gap is not well presented for readers to get the update by the Content and have a structured understanding on up to date information on mitigation with Nosemosis.

Compared with other published material, no addition was achieved. It would be nice to support with info graphs and mode of action details that may inspire applications.

The links are missing with the sections and the conclusion. 

Not well presented using tables and figures. The proposed graphical abstract is not sufficient and relevant. 

The honey bee nosema life cycle has to be presented, and the mode of action-based interactions of the Natural and synthetic substances mentioned in the review in which stage of the life cycle stage has to be presented with references.

Author Response

(The authors gave the same response as above.)

Reviewer 3 Report

The Review "Natural and synthetic substances in the treatment and prevention of honeybee nosemosis" gives a wide view about the research carried out on possible treatments to keep under control Nosemosis, an interesting topic not only for fellow researchers but also for beekeepers.

Authors collected results published in a vast period of time, and recent studies are also cited.

However, to give more significance to the review, I would increase the introduction by underlining also the importance of the disease, by further describing the epidemiological relevance of Nosema spp., pathogenesis and alterations of the disease.

Author Response

Thank you very much for your insightful remarks. Thank you for your time and effort. We made our best to adhere to your suggestions. The latest version of manuscript was proofread by English language author.

The specific gap is not well presented for readers to get the update by the Content and have a structured understanding on up to date information on mitigation with Nosemosis. Compared with other published material, no addition was achieved. It would be nice to support with info graphs and mode of action details that may inspire applications. The links are missing with the sections and the conclusion. 

--We rearranged some paragraph throughout the text to be more specific and easy to follow by readers. We also add some text to be more specific and link sections.

Not well presented using tables and figures. The proposed graphical abstract is not sufficient and relevant. The honey bee nosema life cycle has to be presented, and the mode of action-based interactions of the Natural and synthetic substances mentioned in the review in which stage of the life cycle stage has to be presented with references.

--We added the Nosema spp. life cycle scheme and add information about the mode of action of the natural and synthetic substances. We have changed graphical abstract.

However, to give more significance to the review, I would increase the introduction by underlining also the importance of the disease, by further describing the epidemiological relevance of Nosema spp., pathogenesis and alterations of the disease.

--We have increase the introduction by underlining the importance of nosemosis and describe  pathogenesis and alterations of the disease. We have also added the Nosema spp. life cycle scheme.

--Additionally, we have addressed all specific comments added to the manuscript and discussed with all suggested publication in the field.

We made our best to adhere to your suggestions. After all improvement you have suggested the article is much better and easier to follow by readers.

Thank you once more,

Authors